# In Vitro Screening of Chicken-Derived *Lactobacillus* Strains that Effectively Inhibit *Salmonella* Colonization and Adhesion

**DOI:** 10.3390/foods10030569

**Published:** 2021-03-09

**Authors:** Dan Hai, Zhaoxin Lu, Xianqing Huang, Fengxia Lv, Xiaomei Bie

**Affiliations:** 1College of Food Science and Technology, Nanjing Agricultural University, Nanjing 210095, China; 2017208019@njau.edu.cn (D.H.); fmb@njau.edu.cn (Z.L.); lvfengxia@njau.edu.cn (F.L.); 2College of Food Science and Technology, Henan Agricultural University, Zhengzhou 450000, China; hxq8210@126.com

**Keywords:** chicken-derived *Lactobacillus*, chicken-derived *Salmonella*, bacteriostatic, Caco-2 cells

## Abstract

Inhibition of *Salmonella* by *Lactobacillus* has been a popular research topic for decades; however, the inhibition potential of chicken-derived *Salmonella* by chicken-derived *Lactobacillus* has not yet been studied. In this study, 89 strains of *Lactobacillus* from chicken intestines were isolated by national standard method, Gram staining, physiological, and biochemical experiments and molecular sequencing; The inhibition characteristics of 89 strains of chicken derived *Lactobacillus* against 10 strains *Salmonella* (*S*. Enteritidis SE05, SC31, SC21, SC72 SC74, SC79, SC83, SC87; *S*. bongori SE47; *S*. Typhimurium, SC85) were detected by agar inhibition zone, The results showed that the inhibition zone of 24 strains of chicken derived Lactobacillus was more than 10 mm, which indicated that the isolated chicken derived *Lactobacillus* could effectively inhibit the growth of *Salmonella*; The drug resistance and bile salt tolerance of these 24 strains were analyzed, The results showed that the standard strains LG and L76 were not resistant, and the other 22 *Lactobacillus* strains showed different degrees of resistance. The strains LAB24, LAB26, LAB53, LAB69, and L76 showed good tolerance at the concentration of 3 g/L bile salt; Caco-2 cell experiment and flow cytometry were used to analyze the inhibitory effect of chicken derived *Lactobacillus* on the adhesion of *Salmonella* to Caco-2 cells, The results showed that 16 probiotics could effectively inhibit the adhesion of *Salmonella* to Caco-2 cells. Twelve probiotics were identified by molecular biology. The results showed that L76 was *Enterococcus faecalis*, and the other 11 strains were *Lactobacillus.*

## 1. Introduction

Salmonellosis is a zoonotic food borne disease which causes outbreaks and sporadic cases of gastroenteritis in human worldwide [1]. Chickens have been known as the major source of *Salmonella* contaminated food products such as chicken eggs and meat that cause human salmonellosis in many countries [2,3]. Although poultry are asymptomatic *Salmonella* carriers, and their production performance is unaffected, *Salmonella* can continuously colonize the cecum of broilers [4,5]. During slaughtering and processing, the infected broiler contaminates the meat products, thereby causing food poisoning to humans through the food chain. Infected laying hens can contaminate their eggs, thereby vertically transmitting the infection to the offspring. Salmonellosis may also be caused by ingesting egg products contaminated by *Salmonella*. Notably, in recent years, multiple resistant *Salmonella* strains have been detected. Moreover, the continued use of antibiotics as growth promoters and to control *Salmonella* colonization of broilers may result in more resistant strains. In many Chinese regions, meat products and poultry have been found positive for a variety of foodborne *Salmonella* strains [6,7].

Lactic acid bacteria (LAB) are widely used probiotic organisms, and their strains usually occur in human and animal intestinal microbiota. LAB promote the development of host-favorable microbiota reduce or prevent the colonization of harmful pathogens, enhance mucosal immunity, improve the digestibility of the gastrointestinal tract and reduce its pH, and enhance the maturity and integrity of the intestinal tissue [4]. Additionally, some LAB strains are tolerant to the effects of digestive products, such as gastric acid and bile salts [8], and they can adhere to the host intestinal epithelium [9], thereby gaining a competitive advantage and are important for bacterial maintenance [10]. In vitro models with intestinal cell lines are widely used. Laboratory models using human intestinal cell lines, such as Caco-2 [11], have been developed to study the adhesion of probiotic LAB and the competitive exclusion of pathogenic bacteria.

The aim of this study is to study the efficacy of chicken-derived probiotics on *Salmonella (S*. Enteritidis SE05, SC31, SC21, SC72 SC74, SC79, SC83, SC87; *S.* bongori SE47; *S*. Typhimurium, SC85) by preliminary screening. The isolated strains of chicken-derived probiotics with better antibacterial effect were selected, their surface characteristics, bile salt tolerance and drug resistance of lactic acid bacteria from chicken were studied; The possible reasons of lactic acid bacteria inhibiting *Salmonella* were further analyzed: Study on the adhesion of lactic acid bacteria and the effect of protecting cell membrane.

## 2. Materials and Methods

### 2.1. Strain Isolation, Culture Media, and Used Cell Lines

120 samples of chicken intestines from different slaughterhouses were collected, and *Lactobacillus* were isolated by national standard method [12]. Added 25 g mixed sample to 225 mL normal saline. Mixed at 37 °C for 30 min and serially diluted, coated on de Man, Rogosa, and Sharpe (MRS) agar solid medium, then the coated plates were incubated at 37 °C for 48 h. Typical samples with obvious sour taste were transferred to MRS liquid medium, incubated at 37 °C for 48 h, and then coated again. Then, 107 suspected *Lactobacillus* were selected for Gram staining, and 89 of them turned purple after adding Gram staining agent, microscopic observation showed spherical, short chain, rod-shaped, etc. [13], and then combined with the physiological and biochemical identification of *Lactobacillus,* finally the 89 strains of *Lactobacillus* were isolated. The result show in the Appendix A. *S. Enteritidis* SE05, SC31, SC21, SC72 SC74, SC79, SC83, SC87; *S*. *bongori* SE47; *S*. *Typhimurium*, SC85 were isolated previously [14] and preserved in our laboratory, Caco-2 cells (Shanghai Bogu Biotechnology Co., Ltd., Shanghai, China) were purchased by our laboratory.

### 2.2. Inhibitory Effect of Lactobacillus Strains on Salmonella

The 89 *Lactobacillus* strains were cultured in MRS liquid medium at 37 °C for 48 h, centrifuged at 5000 rpm/min for 5 min, and the supernatant was taken for use. Next, we added 100 μL of *Salmonella* suspension (SE05, SE47, SC21, SC31, SC72, SC74, SC79, SC83, SC85, and SC87) to 100 mL of LB liquid medium, and cultured the mixture at 37 °C and 180 rpm/min for 24 h. Then, 100 mL bottles of solid LB culture medium melted completely and cooled to 40–50 °C, we added 100 μL of culture liquid that contained 10 *Salmonella* isolates, which were cultured overnight; then, the mixture was shaken and poured on the agar plates, which were marked in advance. After the plates cooled, we used a 5 mm punch to drill holes on them. The number of holes drilled depended on the plate size; the discarded agar pieces were picked out with sterilized toothpicks. We added the samples according to the marks on the backs of the plates, as well as 100 μL of LAB supernatant to each hole. Following this, we incubated the plates at 37 °C for 24 h [15]. Twenty-four *Lactobacillus* strains that had the best bacteriostatic effect were selected for subsequent experiments. Three parallel experiments were performed in each group

### 2.3. Antibiotics Resistance of Lactobacillus Strains

Antimicrobial susceptibility of the 24 *Lactobacillus* isolates was evaluated by the disk diffusion test, using Mueller–Hinton agar, according to the guidelines of the Clinical and Laboratory Standards Institute (CLSI, 2012) [16]. The isolates were screened for resistance to the following antibiotics: tetracycline (TET, 30 mg); amoxicillin (AMX, 30 mg); ceftriaxone (CRO, 30 μg); chloramphenicol (CHL, 30 μg); gentamicin (GEN, 10 μg); trimethoprim/sulfamethoxazole (SXT, 1.25/23.75 μg); kanamycin (KAN, 30 μg); erythromycin (ERY, 15 μg). Three parallel experiments were performed in each group. The results were interpreted according to the established CLSI guidelines (CLSI, 2012).

### 2.4. Lactobacillus Bile Salt Tolerance Test

The concentration of bile salt in most animals ranged from 0.3 g/L to 3.0 g/L. in this experiment, the concentration of bile salt was selected in the range of 1.0 g/L–3.0 g/L [17]. The activated bacterial suspension was centrifuged at 4000 rpm for 25 min, and the supernatant was discarded. The precipitated bacteria were washed three times with sterile MRS liquid medium. The bacteria were suspended in the medium, and the bacterial suspension concentration was adjusted to 3.0 × 10^8^ CFU/mL. The reconstituted bacterial suspension was divided into different groups. Next, by adding 1.0, 2.0, and 3.0 g/L of pig and bovine bile salt, the bacterial suspensions were incubated at a constant temperature of 37 °C for 3 h and then removed the supernatant. Gradient dilution was performed immediately by PBS, 1 mL mixture was used to coat the MRS plate. After 48 h of culture at 37 °C, we determined the colony count. The survival rate was calculated as follows: survival rate (%) = number of “2 H” colonies/number of “0 h” colonies × 100% [18]. Three parallel experiments were performed in each group.

### 2.5. Adhesion of Different Bacteria to Caco-2 Cells

Firstly, 1 mL of each *Lactobacillus* or *Salmonella* suspension (2 × 10^8^ CFU/mL) was added to the corresponding 6-well plate containing Caco-2 cells, and cultured at 37 ℃ and 5% CO_2_, Standard *Lactobacillus* strain LG (*L. rhamnosus* GG, FSMM22) as control. Each treatment was repeated three times. After 1 h culture, the culture medium was taken out, the supernatant was discarded, and Caco-2 cells were washed with PBS (pH 7.4) for three times to remove the non-adherent bacteria. The sterile 1% Triton X-100 PBS (1 mL, pH 7.4) solution was added into each well, and it was allowed to stand for 10 min. After the Caco-2 cells were completely removed, the supernatant were transferred into the sterile 1.5 mL centrifuge tube and mixed well. A total of 100 μL of the prepared mixture was diluted at a gradient of 10^−^^1^–10^−^^6^. Then *Lactobacillus* and *Salmonella* were counted on the medium plate [19,20]. The adhesion number of bacteria was calculated using Equation (1):Adhesion index = number of bacteria/cells adhered(1)

### 2.6. Inhibition of Salmonella Adhesion to Caco-2 Cells by Lactobacillus

Caco-2 cells were passaged for 3–4 generations in DMEM complete medium, and then lactic acid bacteria adhesion test was started. The density of Caco-2 cell suspension was adjusted to 2 × 10^5^ cells/mL. A total of 16 strains of *Lactobacillus* were isolated and cultured for 24 h. After centrifugation, the supernatant was removed and the OD value of the suspension was adjusted to 0.6 with PBS. Three *Salmonella* strains were labeled with 1 mL FITC fluorescent solution and incubated in dark for 30 min. The supernatant was removed by centrifugation and resuspended with PBS. A 48 well plate, 200 μL/well was used. First, 48 wells were wetted with 100 μL DMEM solution, then 100 μL cell suspension was added, and 300 μL DMEM incomplete culture medium was added to 500 μL. In the competitive adhesion experiment group, 500 μL LAB + 500 μL fluorescent labeled *Salmonella* were added to each well at the same time; In the rejection adhesion experiment group, 500 μL LAB was added for incubation at 37 °C for 1 h, PBS was washed for 3 times, and then 500 μL fluorescent labeled *Salmonella* for 1 h was added for incubation at 37 °C for 1 h; In the replacement adhesion experiment group, 500 μL fluorescent labeled *Salmonella* was added for 1 h at 37 °C for 1 h, washed with PBS for 3 times, and then 500 μL LAB was added for 1 h at 37 °C for labeling. Finally, all treatments were washed with PBS three times, digested with 0.2 mL trypsin for 2 min, and then added with 0.4 mL DMEM complete culture medium to terminate the reaction. The liquid was collected, and the fluorescence values of each group were detected by the multi-function fluorescent enzyme reader (Infinite M200 Pro multi-function enzyme reader, TECAN, Männedorf, Switzerland). Three parallel experiments were performed in each group.

### 2.7. Effect of Lactobacillus Adhesion to Caco-2 Cells on Their Physiological Metabolism

A total of 1 mL Caco-2 cell suspension (2 × 10^5^ cells/mL) was transferred to a 6-well cell culture plate, and 1 mL DMEM complete culture medium was added and incubated at 37 °C and 5% CO_2_. After the second day of culture, the new DMEM solution was replaced. After the Caco-2 cells completely adhered to the 6-well plate, the used DMEM medium was discarded, and the Caco-2 cells were washed twice with PBS. In each well, 1 mL DMEM medium containing 10% fetal bovine serum was added to a 6-well cell culture plate. The components are as follows: The suspension of CT group (1 mL sterile PBS), LAB group (1 mL 2 × 10^8^ CFU/mL *Lactobacillus*), SAL group (1 mL 2 × 10^8^ CFU/mL *Salmonella*), and rejection group (1 mL of *Lactobacillus* was cultured for 1 h, then 1 mL of *Salmonella* was cultured for 1 h), each group was cultured for 2 h. After culture, the supernatant was collected, and the Caco-2 cells were washed three times with sterile PBS. The Caco-2 cells were dissolved with 1% Triton X-100 (1 mL per well), and the collected lysate stored at -20 °C. Then the alkaline phosphatase (AKP, BC2145, Solarbio, BeiJing, China) [21] and lactate dehydrogenase (LDH, BC0685, Solarbio, BeiJing, China) activities were tested according to the instructions of the sword test box. Three parallel experiments were performed in each group.

### 2.8. Adhesion of Lactobacillus and Salmonella Isolates to Caco-2 Cells and Apoptosis Test

Firstly, 5 mL Cultured Caco-2 cells ((1 − 5) × 10^6^/mL per sample) were directly collected into 10 mL centrifuge tubes and centrifuged at 500–1000 r/min for 5 min. Then, the culture medium was discarded, and the cells were washed with the incubation buffer. Next, they were centrifuged again at 1000 r/min for 5 min. Then, they were labeled, re-suspended in a 100 μL labeling solution, incubated in the dark for 10–15 min at room temperature, centrifuged at 1000 r/min for 5 min, and washed with the incubation buffer. SA-FLOUS (Fluorescent solution) was added, and the mixture was incubated at 4 °C for 20 min, in the dark, with no shaking. Flow cytometry analysis (CytoFLEX LX, BECKMAN, Bria, Florida, USA) showed that the fluorochromes were excited with a 488 nm wavelength. Fluorescein isothiocyanate (FITC) fluorescence was detected with a passband filter at 515 nm wavelength, while PI was detected with a filter at a wavelength greater than 560 nm. Specific parameters of the instrument: Flow pressure: 0–100 psi, step-less adjustable. The liquid flow system can effectively filter 99.999% of the impurities >0.1 μm in the sheath. We used the automatic sample loading system with 1.5 mL sample tubes, which offers the advantage of automatic cleaning, backwashing, bubble removal, and temperature control. Separation speed: 100,000 cells/s, separation purity: >99% (using 70 μm nozzle, at the speed of 70,000 s, 60 psi pressure, the sample with 1% target cell content was separated under the separation mode). The recovery rate: >90% of the theoretically predicted cells. Effective acquisition speed: >100,000 cells/s. Effective sorting speed: >70,000 cells/s. Three parallel experiments were performed in each group.

### 2.9. Purification and Sequencing of PCR Products

Firstly, 12 strains of *Lactobacillus* with typical physiological and biochemical characteristics were screened out by acid-base, bile salt, drug resistance and cell adhesion tests, and then the molecular sequencing test was carried out. 27F(5′-AGAGTTTGATCCTGGCTCAG-3′) and 1492R(5′-GGTTACCTTGTTACGACTT-3′) primers (Primer production, Henan ShangYa Biotechnology Co., Ltd., Zhengzhou, China) were used for PCR amplification and sequencing identification All the PCR products were purified with a High Pure PCR purification kit (Roche, Mannheim, Germany) and sequenced by Nanjing GenScript Biotech Co., Ltd., China. The resultant DNA sequences were analysed using BLAST (http://www.ncbi.nih.gov/BLAST/. accessed date: 19 May 2020).

### 2.10. Statistical Analysis

The experiment was performed triplicate as a replicated test. Data were analyzed using one-way analysis of variance using the SPSS 16.0 software (SPSS Inc., Chicago, Illinois, USA) for Windows. The results are expressed as mean (M) ± standard deviation (SD). Mean separation was performed via Duncan’s multiple range tests (*p* < 0.05).

## 3. Results

### 3.1. Inhibition of Salmonella by Lactobacillus

Figure 1 shows the inhibition results obtained for 10 different *Salmonella* strains by using the bacteriostatic circle method. Different *Lactobacillus* strains had different bacteriostatic effects on different *Salmonella* strains. The diameter of inhibition circle is less than 5 mm, which indicates that the bacteriostatic effect of lactic acid bacteria is not obvious. The diameter of inhibition circle is more than 8 mm, which indicates that lactic acid bacteria have bacteriostatic effect. The diameter of inhibition circle is more than 10 mm, which indicates that the bacteriostatic effect of lactic acid bacteria is obvious. From the Figure 1, the average inhibition zone diameter of the strains LAB1, LAB4, LAB7, LAB16, LAB22, LAB24, LAB35, LAB44, LAB53, LAB54, LAB56, LAB60, LAB64, LAB69, LAB70, LAB73, LAB76, and LG were more than 8 mm and were selected. The inhibition zone of LAB69 (17.2 mm) and LAB76 (17.5 mm) is the largest, indicating that the antibacterial effect is the best.

### 3.2. Antibiotic Resistance of Lactobacillus Strains

The resistance of 24 strains of *Lactobacillus* to antibiotics TET, AMX, CRO, CHL, GEN, SXT, KAN, and ERY were shown in Table 1. The standard strains LG and LAB76 were not drug-resistant. The other 22 strains of LAB showed different degrees of drug resistance. LAB12, LAB14, LAB22, and LAB34 were resistant to 3 antibiotics, 5 strains of lactic acid bacteria were resistant to 2 kinds of antibiotics, and 14 strains were resistant to 1 kind of antibiotics. This may be related to the type and frequency of antibiotics used in poultry breeding. *Lactobacillus* resistant to three kinds of antibiotics were not used in subsequent experiments. Among the 24 strains of *Lactobacillus*, 25% (6/24) were resistant to CRO, 29.2% (7/24) to SXT, 20.8% (5/24) to CHL, and 25% (6/24) to TET. These indicators provide a data base for screening target lactic acid bacteria. 

### 3.3. Lactobacillus Bile Salt Tolerance Test

Figure 2 shows the growth of 24 strains of LAB within 8 h under three different bile salt concentrations. It can be seen from Figure 2 that the growth of 24 *Lactobacillus strains* is not consistent with the increase of bile salt concentration. The OD_600_ value of LAB4, LAB7, LAB24, LAB26, LAB31, LAB35, LAB44, LAB56, LAB65, LAB69, LAB70, L76, and LG were more than 0.20, show that most of the lactic acid bacteria detected in this paper can maintain a high survival rate under the condition of 3 g/L bile salt, and the strains still show good tolerance to the bile salt. At present, many studies have proved that most of the lactic acid bacteria can survive well in the intestine (trypsin, pH 8.0, and 0.3% bile salt), and can return to the initial number of bacteria when passing through the small intestine [22,23].

### 3.4. Bacterial Adhesion to Caco-2 Cells

Adhesion is an important basis for the colonization of *Lactobacillus* in the intestinal tract, and it is necessary to play the role of biological barrier, inhibit the growth and reproduction of pathogenic bacteria, and improve the structure of intestinal flora [24,25]. We screened the *Lactobacillus* with strong adhesion ability from 15 strains of chicken derived *Lactobacillus.*
Figure 3 is the result about the vitro evaluation of the adhesion ability of 15 *Lactobacillus* isolates and 3 *Salmonella* strains using Caco-2 cells as a model. Adhesion index = number of bacteria/cells adhered, the higher the ratio of adhesion index, the stronger the adhesion ability. Fifteen *Lactobacillus* strains had significant differences in their ability to adhere to Caco-2 cells. L76 had the strongest adhesion ability (17.50 bacteria/cell), followed by LAB 69 (15.57 bacteria/cell). The adhesion ability of three *Salmonella* strains was 4.86, 6.03, and 7.57 bacteria/cell, respectively. We selected chicken-derived *Lactobacillus* with higher adhesion ability than *Salmonella* as the target strains of subsequent experiments.

### 3.5. Inhibition of Salmonella Adhesion to Caco-2 Cells by Lactobacillus

The treatment of Caco-2 cells by different bacterial combinations is shown in Table 2. *Salmonella* were incubated with fluorescence for 30 min, and *Lactobacillus* were not treated. The experimental group was incubated with Caco-2 cells according to different combinations. The stronger the fluorescence intensity was, the redder the color was, and the more *Salmonella* adhered to Caco-2 cells. 

On the contrary, the fluorescence intensity was weaker, and the color was blue, indicating that the number of *Salmonella* adhered to Caco-2 cells was less. Adding *Lactobacillus* and *Salmonella* labeled with fluorescence to Caco-2 cells can show the change of fluorescence intensity. The changes of cell adhesion of *Lactobacillus* and *Salmonella* in different treatments were studied by color change. In the competition experiment, *Lactobacillus* and *Salmonella* were added to Caco-2 cells at the same time. The results showed that *Salmonella* in SC31-LAB84, SC79-LAB34 and SC79-LAB72 groups adhered more to Caco-2 cells than other groups (Figure 4a). In the rejection experiment, Caco-2 cells were incubated with *Lactobacillus* for 1 h, and then incubated with *Salmonella* (Figure 4b). The results showed that *Salmonella* had more adhesion to Caco-2 cells in LAB54-SC31 group. In other treatments, *Lactobacillus* adhered more to Caco-2 cells. In the substitution experiment (Figure 4c), Caco-2 cells were incubated with *Salmonella* for 1 h, and then incubated with *Lactobacillus*. The results showed that seven groups of treatments (SC31-LAB35, SC31-LAB44, SC31-LAB53, SC31-LAB54, SC79-LAB24, SC79-LAB34, and SC79-LAB44) had stronger fluorescence intensity, and indicated more *Salmonella* adhesion to Caco-2 cells. While the adhesion rate in other groups was inhibited by *Lactobacillus,* based on the color change.

### 3.6. Lactobacilli Adherence Effect on the Physiological Metabolism of Caco-2 Cells

Caco-2 cells incubated with PBS was the control group. LAB group were incubated with *Lactobacillus,* and the SAL group were incubated with *Salmonella.* The AKP activity of Caco-2 cell lysate and supernatant in the control group were 1378.21 U/gprot and 79.95 U/gprot, respectively (Table 3). LAB group and SAL group (SC31, SC79 and SE05) had different effects on the metabolism of Caco-2 cells. Compared with the control group, the AKP activity of Caco-2 cell supernatant in SAL group was significantly higher than that of Caco-2 cell lysate. SAL significantly inhibited the activity of AKP in Caco-2 cells, destroyed the integrity of cell membrane, and resulted in a large amount of AKP leakage into the cell culture supernatant. Compared with the control group, the AKP activity in the supernatant of Caco-2 cells treated with LAB decreased by 15–59.6% (*p* < 0.01), and the AKP activity in Caco-2 cells treated with LAB also decreased significantly, but it was still 3.45–4.92 times higher than that in SAL group (*p* < 0.01). The AKP activity of Caco-2 cells treated with LAB-SAL was slightly higher than that of SAL, while the AKP content in the supernatant of Caco-2 cells treated with LAB-SAL was significantly lower than that of control group. These results indicate that *Lactobacillus* can repel *Salmonella* and protect Caco-2 cell membrane.

The effects of different treatments on LDH metabolism of Caco-2 cells were significantly different (Table 4). Compared with the control group, LDH activity was not detected in Caco-2 cells incubated with SAL, while LDH activity in the supernatant of Caco-2 cells incubated with SAL increased by 64.73% (*p* < 0.01). It may be that SAL treatment destroys the integrity of Caco-2 cell membrane and releases LDH into cell culture supernatant. The LDH activity in Caco-2 cells incubated with LAB increased significantly (*p* < 0.05), but decreased by 98.49% (*p* < 0.01). The LDH activity of Caco-2 cells incubated with LAB-SAL was significantly higher than that of Caco-2 cells incubated with SAL (*p* < 0.05); the LDH activity content in the supernatant of Caco-2 cells incubated with LAB-SAL increased by 35.26% (*p* < 0.01), but it was still lower than that of SAL. These results suggest that LAB can enhance the LDH metabolic activity of Caco-2 cells, which further confirms the protective effect of SAL on the integrity of Caco-2 cell membrane by preventing the destruction of Caco-2 cell membrane.

### 3.7. Lactobacillus and Salmonella Isolate Adhesion to Caco-2 Cells and Apoptosis Test Results

Apoptosis was detected using flow cytometry (CytoFLEX LX, BECKMAN, USA). Propidium iodide (PI) is a type of nucleic acid dye. Under normal circumstances, PI cannot penetrate the entire cell membrane; this is not true for mid- and late apoptotic as well as dead cells, the cell membrane of which PI penetrates and dyes the nucleus red. Caco-2 cells treated with PBS were used as blank control group, multidrug resistant *Salmonella* SC79 was used as negative control group, and standard *Lactobacillus* strain LG was used as positive control group. Figure 5 shows the four quadrant method to observe the apoptosis of Caco-2 cells, that is, the scatter plot of bivariate flow cytometry. In the Figure 5, P1 gate showed the ratio of living cells, which shows the number of living cells detected by flow cytometry after Caco-2 cells were incubated with different bacteria. Compared with the blank control group, the number of Caco-2 cells in LAB69 group and LG group were significantly increased, while the number of Caco-2 cells in LAB69 group was higher than that in the positive control group; the number of Caco-2 cells in the negative control group was significantly decreased. P2 gate and P3 gate show the apoptotic and dead cells detected by flow cytometry after Caco-2 cells were incubated with different bacteria. Compared with the blank control group, the number of apoptotic cells and dead cells of Caco-2 cells in LAB69 group and LG group decreased, and the number of apoptotic cells and dead cells in LAB69 group was lower than that in the positive control group; while the number of apoptotic cells and dead cells in Caco-2 cells in SC79 group increased significantly. As can be seen from Figure 5, compared with the blank control group, the results of Caco-2 cells incubated with different bacteria were significantly different. The invasion and destruction of Caco-2 cells in *Salmonella* negative control group were more serious, Compared with positive control group LG and LAB69, the number of live cells increased and the number of dead cells decreased. These results indicate that LAB69 has a certain protective effect on Caco-2.

### 3.8. Molecular Identification of Lactobacillus Strains

After PCR amplification, electrophoresis results of Lactobacillus amplification products was show in the Appendix A. The sequencing results in Figure 6 and Table 5. In this experiment, 12 strains of chicken derived lactic acid bacteria with good inhibitory effect on *Salmonella*, weak drug resistance, strong bile salt resistance, and strong adhesion to Caco-2 cells were selected for molecular sequencing analysis, according to the experimental results in Table 5, LAB53, LAB60, and LAB64 are the same strain; that is, *Lactobacillus salinus* strain. Except *Lactobacillus reuteri* strain LAB4 and *Enterococcus faecium* strain IAH L76, the rest were *Lactobacillus plantarum* strains.

## 4. Discussion

In this experiment, 89 strains of lactic acid bacteria were isolated from Chicken Intestines of different slaughterhouses according to the national standard method and Gram staining. Using *Salmonella* as indicator bacteria and agar diffusion inhibition zone method, 24 strains of chicken-derived lactic acid bacteria with an average inhibition diameter of more than 10 mm were preliminarily screened. Due to the cost, all 24 strains of chicken-derived lactic acid bacteria were not identified by molecular sequencing. The drug resistance of these 24 strains of lactic acid bacteria was analyzed. Except for LAB 76 and LG, the drug resistance rate of 24 strains of lactic acid bacteria was 33%, gentamicin 21%, chloramphenicol 21%, ceftriaxone sodium 25% and tetracycline 25%. LAB12, LAB22, and LAB34 were resistant to three kinds of antibiotics, so the selected lactic acid bacteria were selected from the remaining 21 strains. At the same time, the bile salt tolerance of 24 lactic acid bacteria isolates was detected, the OD600 value of LAB4, LAB7, LAB24, LAB26, LAB31, LAB35, LAB44, LAB56, LAB65, LAB69, LAB70, LAB76, and LG were more than 0.20, and the strains still show good tolerance to the bile salt. The cell adhesion ability of 15 LAB strains differed depending on the species and strain. Murphy (2009) [26] also confirmed great adhesion differences among different strains and even within the same strain. Thus, adhesion ability is suggested to have an inevitable relationship with LAB characteristics. Adhesion can be non-specific (related to surface properties) or specific (related to adhesives) [27]. The existing adhesives include surface proteins, pili, peptidoglycans, and lipopolysaccharides. Studies have shown that the S-layer protein on the surface of *Lactobacillus* strains may be the active adhesion site [28]. Based on the aforementioned considerations, the adhesion of Lactobacilli to Caco-2 cells may result from the orchestrated action of various adhesives. The mechanism of *Lactobacillus* antagonizing the adhesion of *Salmonella* to Caco-2 cells was explored by three ways of exclusion, competition and substitution; the protective effect of *Lactobacillus* on intestinal cells was preliminarily analyzed by flow cytometry to detect the permeability of cell membrane; and the biological function of *Lactobacillus* SLP was preliminarily explored These results suggest that LAB group bacteria protected the AKP metabolism activity and Caco-2 cell membrane integrity by reducing the damage caused by the SAL group to the Caco-2 cell membrane.

Our results showed that the bacteriostatic zones of LAB69 and L76 were the widest (17.2 mm and 17.5 mm, respectively). Additionally, most of the 24 *Lactobacillus* strains inhibited the adhesion of *Salmonella* to the Caco-2 cell surface bacteria effectively. Moreover, after attaching to the cell surface, *Lactobacillus* effectively controlled the reattachment of *Salmonella* bacteria to cells. Further, 16 *Lactobacillus* strains could replace *Salmonella* strains effectively and adhere to the surface of Caco-2 cells. In general, incubation with *Lactobacillus* had protective effects on Caco-2 cell membranes, and the detection results were close to that of the control group.

## 5. Conclusions

In this experiment, we isolated chicken-derived *Lactobacillus*, which can effectively inhibit the colonization and adhesion of *Salmonella*, from chicken intestines. The LAB53, LAB60, LAB69, LAB72, and LAB76 *Lactobacillus* isolates had slightly higher tolerance and adhesion capacity, as well as probiotic potential than the standard LG strain; however, the probiotic effect of *Lactobacillus* isolates on the host after adhering to the intestinal tract should be studied further. We speculate that the incorporation of chicken-derived LAB during the later stage of chicken feeding might curtail *Salmonella* outbreaks in the breeding and production processes of chicken eggs, ultimately improving food safety, Additionally, the LAB69 and LAB76 isolates had good adhesion ability and may be rich in surface proteins. Their adhesion properties and mechanism need to be studied in detail in the future.

## Figures and Tables

**Figure 1 foods-10-00569-f001:**
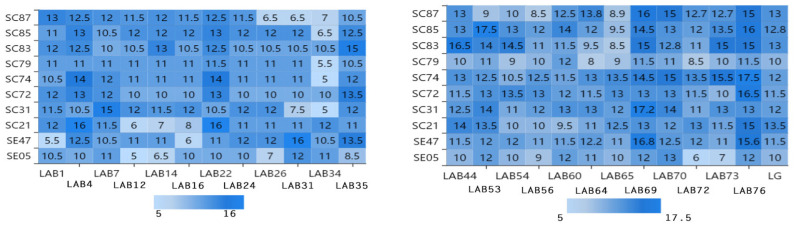
Experimental results of inhibitory zone of 24 *Lactobacillus* isolates to 10 *Salmonella* isolates. Note: The vertical axis is 10 strains of *Salmonella*, the horizontal axis is 24 strains of *Lactobacillus*, and the intersection of the horizontal axis and the vertical axis is the diameter of inhibition zone of *Lactobacillus* inhibiting *Salmonella*. (The diameter of inhibition zone: mm).

**Figure 2 foods-10-00569-f002:**
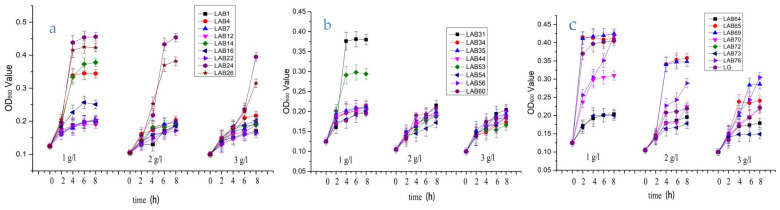
The bile salt tolerance of 24 strains of *Lactobacillus*. Note: (**a**)-The growth of 8 strains of *Lactobacillus (*LAB1, LAB4, LAB7, LAB12, LAB14, LAB16, LAB22, LAB24) under the condition of 1 g/L, 2 g/L, 3 g/L bile salt concentration for 8 h; (**b**)-The growth of 9 strains of *Lactobacillus (*LAB26, LAB31, LAB34, LAB35, LAB44, LAB53, LAB54, LAB56,LAB60) under the condition of 1 g/L, 2 g/L, 3 g/L bile salt concentration for 8 h; (**c**)-The growth of 8 strains of *Lactobacillus (*LAB64, LAB65, LAB69, LAB70, LAB72, LAB73, LAB76, LG) under the condition of 1 g/L, 2 g/L, 3 g/L bile salt concentration for 8 h.

**Figure 3 foods-10-00569-f003:**
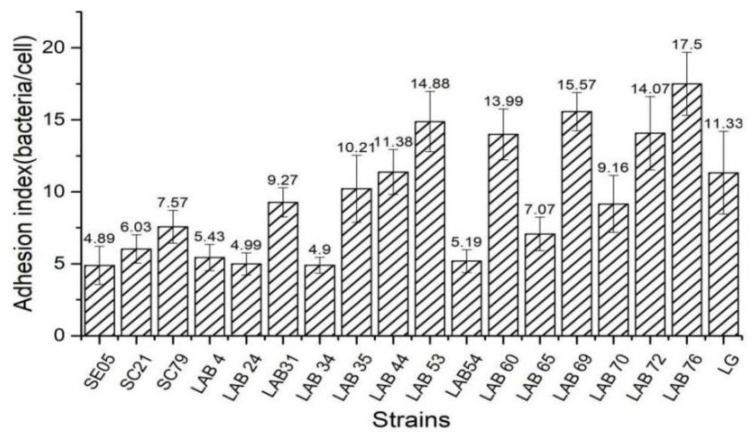
Adhesion index of different bacteria to CaCO-2 cells.

**Figure 4 foods-10-00569-f004:**
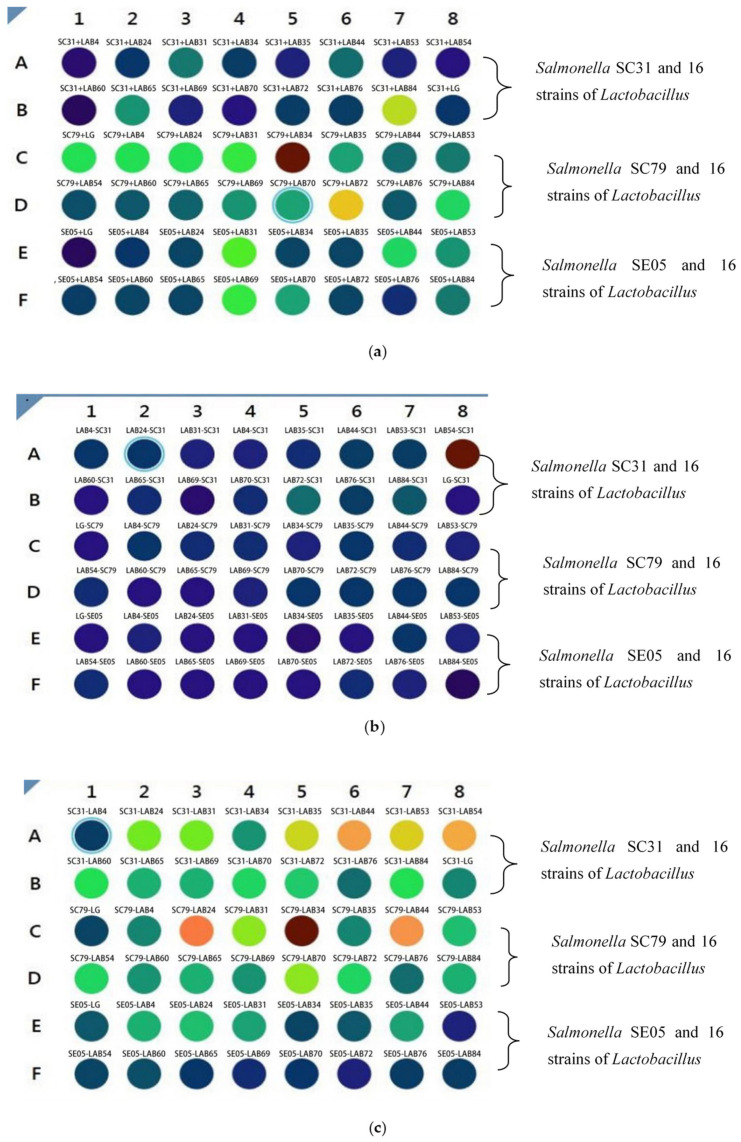
(**a**) Competitive adhesion test of Caco-2 cells incubated with *Lactobacillus* and *Salmonella.* In Figure 4a, The target *Salmonella* are SC31, SC85, and SE05. In this experiment, *Lactobacillus* and *Salmonella* were added at the same time and incubated for 1 h, then the fluorescence intensity was measured. The stronger the fluorescence intensity was, the more red the color was. It showed that the number of *Salmonella* incubated on Caco-2 cells was more, while the fluorescence intensity was weak, the color was blue, indicating that the number of *Salmonella* incubated on Caco-2 cells was small. (**b**) Rejection and adhesion test of Caco-2 cells incubated by lactic acid bacteria first and then by Salmonella. In Figure 4b, the *Lactobacillus* isolated strains were added for incubation for 1 h, then the target *Salmonella* SC21, SC85 and SE05 were added for incubation for 1 h, and the results were measured by multi-functional enzyme labeling instrument. The stronger the fluorescence intensity was, the more red the color was. It showed that the number of *Salmonella* incubated on Caco-2 cells was more, while the fluorescence intensity was weak, the color was blue, indicating that the number of *Salmonella* incubated on Caco-2 cells was small. (**c**) Displacement adhesion test of Caco-2 cells incubated by *Salmonella* and then by lactic acid bacteria. In Figure 4c, the target *Salmonella* SC21, SC85 and SE05 were added to incubate for 1 h, and then the *Lactobacillus* isolate was added to incubate for 1 h, and then the results were measured by multi-functional enzyme labeling instrument. The stronger the fluorescence intensity was, the more red the color was. It showed that the number of *Salmonella* incubated on Caco-2 cells was more, while the fluorescence intensity was weak, the color was blue, indicating that the number of *Salmonella* incubated on Caco-2 cells was small.

**Figure 5 foods-10-00569-f005:**
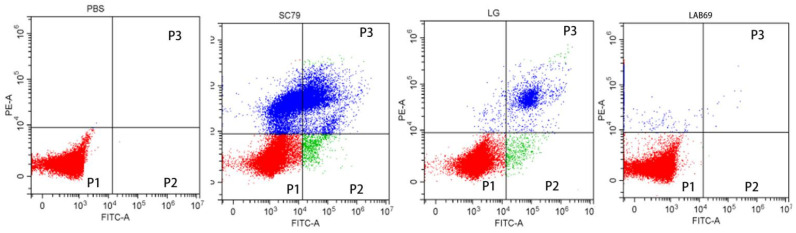
Four quadrants results of Caco-2 cells treated with different bacteria in flow cytometry. Note: P1 gate, the red part of P1 gate is the ratio of the number of living cells. The higher the red ratio, the more living Caco-2 cells; P2 gate, the green part of P2 gate is the ratio of apoptotic cells. The larger the green area, the more apoptotic cells; P3 gate, The blue part of P3 gate is the ratio of death cells. The larger the blue area, the more death cells.

**Figure 6 foods-10-00569-f006:**
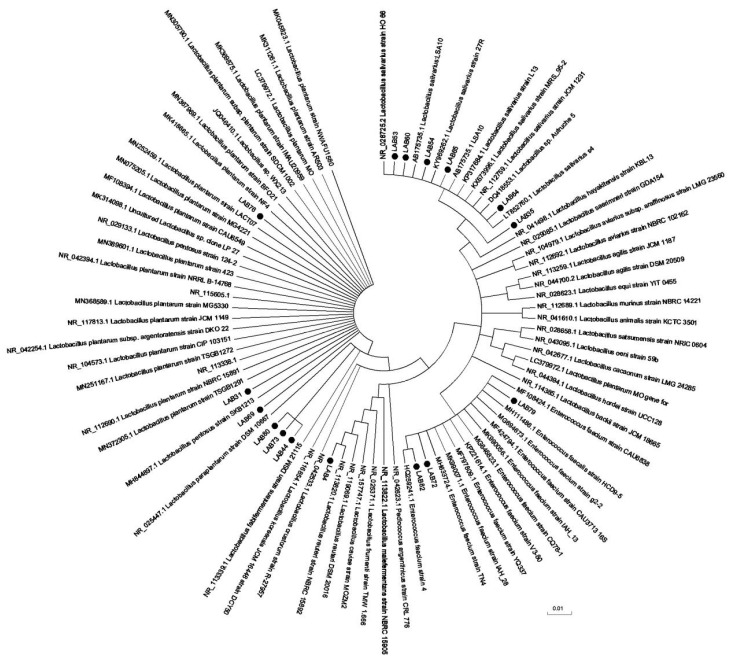
Neighbor-joining phylogenetic tree based on 16 S rRNA gene sequences showing relationship between *Lactobacillus* isolates.

**Table 1 foods-10-00569-t001:** Drug resistance of 24 strains of lactic acid bacteria.

Strains Antibiotic	LAB	LAB	LAB	LAB	LAB	LAB	LAB	LAB	LAB	LAB	LAB	LAB	
	1	4	7	12	14	16	22	24	26	31	34	35	
GEN	S	S	S	S	R	S	R	R	S	S	S	S	
KAN	S	S	S	R	S	S	S	S	S	S	R	S	
AMX	S	S	S	R	S	R	S	S	S	S	S	S	
CRO	R	R	S	S	R	S	R	S	R	S	S	S	
SXT	S	S	R	S	S	R	S	S	S	S	R	S	
CHL	R	S	S	S	R	S	S	S	S	R	S	S	
TET	S	R	S	R	S	S	R	S	S	S	S	R	
ERY	S	S	S	S	S	S	S	S	R	S	R	S	
Total number	2	2	1	3	3	2	3	1	2	1	3	1	
	LAB	LAB	LAB	LAB	LAB	LABV	LAB	LAB	LAB	LAB	LAB	LAB	
GEN	44	53	54	56	60	64	65	69	70	72	73	76	LG
KAN	S	S	S	S	S	R	S	S	R	S	S	S	S
AMX	S	S	S	S	S	S	S	S	S	S	S	S	S
CRO	S	S	S	S	S	S	S	S	S	S	S	S	S
SXT	S	S	S	S	S	S	S	S	S	S	R	S	S
CHL	S	R	R	S	R	S	S	S	S	R	S	S	S
TET	S	S	S	R	S	S	R	S	S	S	S	S	S
ERY	R	S	S	S	S	S	S	R	S	S	S	S	S
Total number	S	S	S	S	S	S	S	S	S	S	R	S	S
	1	1	1	1	1	1	1	1	1	1	2	0	0

Note: S (susceptible), R (Resistant).

**Table 2 foods-10-00569-t002:** Treatment of Caco-2 cells with different combinations of bacteria.

Single Bacteria Group	Competitive Treatment Group	Rejection Treatment Group	Replacement Treatment Group
SC79 (*n* = 3)	LAB44 + SC79 (*n* = 3)	LAB44—SC79 (*n* = 3)	SC79—LAB44 (*n* = 3)
LAB35 (*n* = 3)	LAB53 + SC79 (*n* = 3)	LAB53—SC79 (*n* = 3)	SC79—LAB53 (*n* = 3)
LAB44 (*n* = 3)	LAB35 + SC79 (*n* = 3)	LAB35—SC79 (*n* = 3)	SC79—LAB35 (*n* = 3)
LAB53 (*n* = 3)	LAB69 + SC79 (*n* = 3)	LAB69—SC79 (*n* = 3)	SC79—LAB69 (*n* = 3)
LAB69 (*n* = 3)	LAB76 + SC79 (*n* = 3)	LAB76—SC79 (*n* = 3)	SC79—LAB76 (*n* = 3)
LAB76 (*n* = 3)	LG + SC79 (*n* = 3)	LG—SC79 (*n* = 3)	SC79—LG (*n* = 3)
LG (*n* = 3)			

**Table 3 foods-10-00569-t003:** Activity of alkaline phosphatase (AKP) in lysate and supernatant of Caco-2 cells incubated with 12 strains of *Lactobacillus* and 3 strains of *Salmonella.*

Treatment Group	Cell Culture Supernatant AKP (U/100 mL)	Cell Lysate AKP (U/gprot)	Treatment Group	Cell Culture Supernatant AKP (U/100 mL)	Cell Lysate AKP (U/gprot)
CT(PBS)	1378.21 ± 1.36	79.95 ± 1.36	LAB76-SC31	28.56 ± 1.36	178.25 ± 1.36
LAB24	321.97 ± 1.36	29.66 ± 1.36	LG-SC31	29.37 ± 1.36	167.91 ± 1.36
LAB31	391.54 ± 1.36	18.44 ± 1.36	LAB24-SC79	39.97 ± 1.36	152.09 ± 1.36
LAB35	401.62 ± 1.36	14.25 ± 1.36	LAB31-SC79	33.56 ± 1.36	161.21 ± 1.36
LAB44	339.61 ± 1.36	47.67 ± 1.36	LAB35-SC79	29.77 ± 1.36	179.74 ± 1.36
LAB53	413.39 ± 1.36	14.09 ± 1.36	LAB44-SC79	40.92 ± 1.36	146.34 ± 1.36
LAB60	400.83 ± 1.36	12.93 ± 1.36	LAB53-SC79	27.42 ± 1.36	173.99 ± 1.36
LAB65	319.57 ± 1.36	34.72 ± 1.36	LAB60-SC79	33.88 ± 1.36	166.83 ± 1.36
LAB69	414.06 ± 1.36	13.55 ± 1.36	LAB65-SC79	49.37 ± 1.36	130.65 ± 1.36
LAB70	399.51 ± 1.36	15.78 ± 1.36	LAB69-SC79	21.89 ± 1.36	174.64 ± 1.36
LAB72	344.21 ± 1.36	30.19 ± 1.36	LAB70-SC79	39.77 ± 1.36	159.82 ± 1.36
LAB76	423.01 ± 1.36	10.68 ± 1.36	LAB72-SC79	40.03 ± 1.36	153.53 ± 1.36
LG	405.3 ± 1.36	17.83 ± 1.36	LAB76-SC79	23.56 ± 1.36	175.11 ± 1.36
SC31	92.04 ± 1.36	133.77 ± 1.36	LG-SC79	28.64 ± 1.36	169.48 ± 1.36
SC79	87.21 ± 1.36	156.22 ± 1.36	LAB24-SE05	51.77 ± 1.36	138.79 ± 1.36
SE05	86.49 ± 1.36	163.06 ± 1.36	LAB31-SE05	47.32 ± 1.36	156.05 ± 1.36
LAB24-SC31	47.42 ± 1.36	143.35 ± 1.36	LAB35-SE05	18.88 ± 1.36	176.43 ± 1.36
LAB31-SC31	41.32 ± 1.36	150.65 ± 1.36	LAB44-SE05	52.73 ± 1.36	139.03 ± 1.36
LAB35-SC31	39.88 ± 1.36	169.44 ± 1.36	LAB53-SE05	37.11 ± 1.36	159.04 ± 1.36
LAB44-SC31	40.77 ± 1.36	146.59 ± 1.36	LAB60-SE05	37.28 ± 1.36	164.03 ± 1.36
LAB53-SC31	37.97 ± 1.36	153.29 ± 1.36	LAB65-SE05	55.42 ± 1.36	133.57 ± 1.36
LAB60-SC31	40.32 ± 1.36	156.35 ± 1.36	LAB69-SE05	21.32 ± 1.36	170.65 ± 1.36
LAB65-SC31	59.77 ± 1.36	139.74 ± 1.36	LAB70-SE05	29.48 ± 1.36	163.48 ± 1.36
LAB69-SC31	30.56 ± 1.36	171.21 ± 1.36	LAB72-SE05	46.83 ± 1.36	159.52 ± 1.36
LAB70-SC31	35.88 ± 1.36	166.43 ± 1.36	LAB76-SE05	20.36 ± 1.36	173.05 ± 1.36
LAB72-SC31	43.73 ± 1.36	149.53 ± 1.36	LG-SE05	22.63 ± 1.36	168.57 ± 1.36

**Table 4 foods-10-00569-t004:** Lactate dehydrogenase (LDH) activity in lysate and supernatant of Caco-2 cells incubated with 12 strains of *Lactobacillus* and 3 strains of *Salmonella.*

Treatment Group	Cell Culture Supernatant LDH (U/L)	Cell Lysate LDH (U/gprot)	Treatment Group	Cell Culture Supernatant LDH (U/L)	Cell Lysate LDH (U/gprot)
CT(PBS)	0 ± 0.22	0 ± 0.22	LAB24-SC79	577.32 ± 0.22	30.66 ± 0.22
LAB24	23.1 ± 0.22	18.76 ± 0.22	LAB31-SC79	596.38 ± 0.22	32.54 ± 0.22
LAB31	24.58 ± 0.22	11.27 ± 0.22	LAB35-SC79	561.88 ± 0.22	40.33 ± 0.22
LAB35	39.87 ± 0.22	30.5 ± 0.22	LAB44-SC79	592.37 ± 0.22	21.46 ± 0.22
LAB44	46.25 ± 0.22	31.84 ± 0.22	LAB53-SC79	560.34 ± 0.22	39.27 ± 0.22
LAB53	35.57 ± 0.22	27.09 ± 0.22	LAB60-SC79	603.66 ± 0.22	21.88 ± 0.22
LAB60	40.09 ± 0.22	31.59 ± 0.22	LAB65-SC79	705.39 ± 0.22	18.41 ± 0.22
LAB65	27.89 ± 0.22	17.51 ± 0.22	LAB69-SC79	548.17 ± 0.22	43.67 ± 0.22
LAB69	42.55 ± 0.22	34.27 ± 0.22	LAB70-SC79	613.05 ± 0.22	23.49 ± 0.22
LAB70	33.59 ± 0.22	29.66 ± 0.22	LAB72-SC79	578.02 ± 0.22	28.44 ± 0.22
LAB72	31.67 ± 0.22	21.55 ± 0.22	LAB76-SC79	559.55 ± 0.22	41.81 ± 0.22
LAB76	47.66 ± 0.22	38.78 ± 0.22	LG-SC79	573.22 ± 0.22	36.19 ± 0.22
LG	41.92 ± 0.22	33.76 ± 0.22	LAB24-SE05	651.44 ± 0.22	19.31 ± 0.22
SC31	913.5 ± 0.22	0 ± 0.22	LAB31-SE05	600.23 ± 0.22	23.37 ± 0.22
SC79	885.76 ± 0.22	0 ± 0.22	LAB35-SE05	552.19 ± 0.22	40.55 ± 0.22
SE05	848.27 ± 0.22	0 ± 0.22	LAB44-SE05	636.15 ± 0.22	20.94 ± 0.22
LAB24-SC31	611.22 ± 0.22	19.36 ± 0.22	LAB53-SE05	549.92 ± 0.22	44.52 ± 0.22
LAB31-SC31	605 ± 0.22	20.41 ± 0.22	LAB60-SE05	579.11 ± 0.22	24.68 ± 0.22
LAB35-SC31	598.17 ± 0.22	22.57 ± 0.22	LAB65-SE05	644.24 ± 0.22	19.35 ± 0.22
LAB44-SC31	573.05 ± 0.22	27.49 ± 0.22	LAB69-SE05	590.03 ± 0.22	29.44 ± 0.22
LAB53-SC31	571.55 ± 0.22	33.88 ± 0.22	LAB70-SE05	588.64 ± 0.22	30.39 ± 0.22
LAB60-SC31	608.02 ± 0.22	21.46 ± 0.22	LAB72-SE05	599.65 ± 0.22	27.56 ± 0.22
LAB65-SC31	631.42 ± 0.22	20.31 ± 0.22	LAB76-SE05	553.09 ± 0.22	46.43 ± 0.22
LAB69-SC31	561.09 ± 0.22	32.65 ± 0.22	LG-SE05	564.57 ± 0.22	41.82 ± 0.22
LAB70-SC31	599.14 ± 0.22	21.59 ± 0.22	LAB76-SE05	553.09 ± 0.22	46.43 ± 0.22
LAB72-SC31	578.06 ± 0.22	20.58 ± 0.22	LG-SE05	564.57 ± 0.22	41.82 ± 0.22

**Table 5 foods-10-00569-t005:** 16S rDNA sequence analysis results of BLAST.

Strains	Gen Bank Homologous Sequence	The Highest Homologous Strain	Maximum Homology (%)
LAB4	NR_113820.1	*Lactobacillus reuteri* strain NBRC 15892	99.5%
LAB31	NR_112690.1	*Lactobacillus plantarum* strain NBRC 15891	99.9%
LAB35	NR_028725.	*Lactobacillus salinus* strain HO 66	99.8%
LAB44	MF108394.1	*Lactobacillus plantarum* strain cau6549	99.4%
LAB53	NR_112759.1	*Lactobacillus salinus* strain JCM 1231	98%
LAB60	NR_112759.1	*Lactobacillus salinus* strain JCM 1231	99%
LAB64	NR_112759.1	*Lactobacillus salinus* strain JCM 1231	98%
LAB65	KP317684.1	*Lactobacillus salinus* strain L13	99.5%.
LAB69	JQ046410.1	*Lactobacillus* sp. wx213	98%
LAB73	MK311261.1	*Lactobacillus plantarum* strain AR503	99.1%
L76	MK990071.1	*Enterococcus faecium* strain IAH_28	98.20%
LAB79	MN389601.1	*Lactobacillus plantarum* strain 42	98.5%

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
