# Peer review of "In Vitro Screening of Chicken-Derived Lactobacillus Strains that Effectively Inhibit Salmonella Colonization and Adhesion"

_foods, 2021, doi:10.3390/foods10030569_

Round 1

Reviewer 1 Report

The manuscript quality has been substantially improved after the first revision. However there are some aspects to be refine:

L42. In the first revision I mentioned that the term of microflorae is seriously wrong in the 21th Century. Please replace microflorae by microbiota!

Around in text I also mentioned that personal writing is not approved for scientific papers... See point 2.2 of materials and methods and replace for impersonal writing.

In Discussion, replace chicken derive by chicken-derived as you do in conclusions.

L457. Remove the phase complete of "This may be the reason why ...[] ...receive antibiotics"

L462. Remove ""in other words, 13 strains... []... 3g/L.

Both phrases and highly redundant and not offer nothing significantly.

Author Response

Response to Reviewer 1 Comments

The manuscript quality has been substantially improved after the first revision. However there are some aspects to be refine: 

Point 1: L42. In the first revision I mentioned that the term of microflorae is seriously wrong in the 21th Century. Please replace microflorae by microbiota! Around in text I also mentioned that personal writing is not approved for scientific papers... See point 2.2 of materials and methods, and replace for impersonal writing. In Discussion, replace chicken derive by chicken-derived as you do in conclusions.

Response 1: Thanks for your suggestions and we are apologized for this error. This problem may be because poor English didn't make it clear in the previous article. It has been modified in the article.I have changed the microflorae by microbiota and  rewrited the manuscript eliminating all personal forms "we". I have replaced the  chicken derive by chicken-derived.

Point 2: L457.Remove the phase complete of "This may be the reason why ...[] ...receive antibiotics"

Response 2Thanks for your suggestions and we are apologized for this error. I've made changes here in the article at the line 457.

Point 3: L462. Remove ""in other words, 13 strains... []... 3g/L. Both phrases and highly redundant and not offer nothing significantly.

Response 3: Thanks for your suggestions and we are apologized for this error. I've made changes here in the article at the line 462.

Reviewer 2 Report

The authors still need to explain their results then refer to the figurs, not the opposit

Author Response

Response 1: Thanks for your suggestions and we are apologized for these errors. According to your suggestion, we analyzed and explained the data in line 119-219.

This manuscript is a resubmission of an earlier submission. The following is a list of the peer review reports and author responses from that submission.

Round 1

Reviewer 1 Report

General comments

The manuscript by Hai et al. reports the use of Lactobacillus strains as a measure to control Salmonella colonization using the Caco-2 cells in vitro model. Authors conclude that sixteen strains were able to inhibit Salmonella colonization and adhesion to Caco-2 cells. However, with the techniques used and the results exposed those conclusions are overestimated.

Although the manuscript could be interesting due to the application of probiotic strains for avoiding the colonization ability of a critical foodborne pathogen as Salmonella, the manuscript has serious drawbacks. Material and methods section must be rewritten appropriately, impersonally, and describing the methods adequately. For example, it is not detailed the selection criteria for the different strains selected to be applied in the different experiments. Similarly, the cut-off criteria for the different experiments performed. For example, the scales of inhibition diameters for classifying the effect (Point 2.2), the selection of antibiotics for MIC determination of lactic bacteria (point 2.3). Moreover, the methods for determining the adhesion assays of Salmonella and Lactobacillus in Caco cells (points 2.5 and 2.6) need revision and a correct explanation. Similarly, the studies of adherence carried out by flow cytometry must be revised and explain why only Salmonella cells were labeled and not also of Lactobacillus, for example. Use of controls as Salmonella without exposition to Lactobacillus must be included in all these experiments for comparison.

The materials and methods and the results are not rigorously exposed, which makes it practically impossible to extract the information necessary to be able to assess the significance of the results exposed. The results section needs revision and deep analysis, they are explained mostly in a qualitative manner. In most cases, the significance of the results is not included. Moreover, the discussion is scarce superficial, repeating the results and referring in its main part to methods not described in the manuscript. Conclusions are not justified by the data shown.

The abstract must be rewritten appropriately after manuscript revision to show the most outstanding results and conclusions.

Minor comments

The selection of the different strains used in the different experiments must be justified.

  1. Line 62. MRS instead Mrs.
  2. Line 84: What is the meaning of “regulating bacteria”.
  3. Line 117: Please, delete: “The living bacteria in the colony were examined morphologically with a microscope to confirm that the colony was not contaminated”.
  4. Line 148: Please, revise and correct: “Alkaline 147 phosphatase (AKP) and lactate dehydrogenase (LDH) activities were measured at -20 °C”. Is the testing temperature correct?
  5. Line 172: Please, revise as 16 strains are indicated and results of 12 are only shown (Lines 284-296). What was the criteria for selecting those strains?
  6. Line 176: PCR for sequencing of 16S gene, was carried out by real-time PCR?
  7. Line 185: Please, write Salmonella in the italic letter, and revise the manuscript to correct other similar mistakes.
  8. Line 187: Values for the largest inhibition zones for the detailed strains in the text do not agree with those shown in Figure 1. Please, revise and correct. Moreover, the reference of the criteria to determine the cut-off values must be included.
  9. Line 195: If antibiotics are used in farms of the study, this must be indicated. Besides, the criteria for the selection of the antibiotic tested.
  10. Figure 2: Please, detail de units showed in the diagram of this figure and the adhesion index as the bacterial CFU must be referred to ml of cells and not Caco-2 cells. OD values are not appropriate to show the viability of cells. Determination of CFU/mL would be more suitable.
  11. Lines 203-206: Please, revise Figure 2 and the results exposed. There are other strains that seem to have a higher tolerance than exposed.
  12. Line 203, please rewrite, the correct is Figure 2.
  13. Lines 208-212: Please, indicate the cut-off criteria to determine the levels of bacterial cells adhesion.
  14. Line 235: Please, write in a positive sense: “Compared with the LAB group, the SAL groups did not have significantly lower AKP and LDH activities”.
  15. Line 237: This sentence has no sense, please, revise: “Compared with the SAL group, the SAL groups had significantly higher AKP and LDH activities”.
  16. Please, rewrite the text of all the figure legends in a more appropriate and understandable manner
  17. Figure 4.1, can be deleted. The color code can be explained in the legend of the results of figure 4.
  18. Line 215-216: Delete the sentence related to figure 4.1 it is not adequate.
  19. Lines 249-259: Please, delete. This paragraph is similar to the paragraph of lines 226-238. The same for paragraph (lines 260-269 and 239-248).
  20. Figures 4: Please, explain why apoptosis experiments were only performed with one strain of Salmonella and one of Lactobacillus.
  21. Figures 5 legends. Please, rewrite and add the information for interpreting the results shown in diagrams. Lines 273: What does it mean the sentence: “the number of living cells incubated in SC79 273 increased significantly” when it is supposed that Salmonella has an apoptotic effect in Caco-2 cells?
  22. Point 2.6 and Lines 561-567: please, revise as the effect of the application of LiCl and sodium periodate is not described in the Material and Methods section so their effect cannot be discussed and concluded.
  23. Line 281: Please, revise the sentence “According to Figures 5-1, 5-2 and 5-3, the apoptosis of different Lactobacillus and The number of living cells in the 281 LAB69 and LG groups was higher than that in the PBS and SAL groups” and rewrite.
  24. Lines 284-296: Please, delete this paragraph and show the results as a table. If the strains identified as belonging to the same species are the same strain or are clonally related must be indicated.
  25. The size of the letters and situation of Figure 4 in the PDF did not allow the revision of the figure content.

Reviewer 2 Report

The manuscript describes the isolation, characterization and identification of LAB from chicken intestines able to inhibit Salmonella strains as well as their bile salt and antibiotic resistance. Authors also studied in vitro approach by using Caco2 cells lines.

Major revisions:

The overall structure of manuscript is acceptable but english language must be improve in all text. L. 69 Authors must avoid personality in their descriptions therefore they must rewrite the manuscript eliminating all personal forms "we"

How it is possible that through the morphological study carried out with a microscope, the authors describe classic forms of lactobacillus and include them the strain L76 corresponding to enterococcus, whose shape is not characteristic of lactobacilli??

L.173 Sequence of primers 27F and 1492r must be added to the manuscript or references about them. Authors must describe which region is amplify using those primers. Other point is that authors must describe in text how from real-time PCR get the amplified region and the sequencing process in order to obtain the results mentioned in L. 284.

Authors did not mentioned nothing about identification of LAB either in discussion or conclusion...

There are methodologies and affirmation in text without reference to support them. Authors must complete this request.

Minor revisions:

L.42 Replace "Microflorae" or "microflora" by Microbiota please.

L.45 Not all LAB strains are tolerant to the effects of "digestive products", maybe authors is referred to probiotic LAB strains. Modify please.

L. 47 Erase "As the study of in vivo bacterial adhesion is generally difficult".

L. 53 replace "then the" by their

L. 58 what authors' meaning with "mixed treatment"? explain in text.

L. 62 MRS

L. 66 in last paper and preserved in our laboratory must be replaced by previously

L. 72. "After the content of 10" rewrite 

L. 81 replace drug by antibiotics

L. 82. Authors must mention in text the antibiotics used and their concentration range.  

L. 152 to 156. That description must be replace to results section.

Authors must reduce figures number or replace as supplementary results. Figure 4-1, 5-1 and 5-4 must be erase or replace to supplementary.

Figures 5-2 and 5-3 can be published together as only one.

L. 388 "in figure 14" must be rewrite.

L. 587 " the identity of Lactobacillus protein" what is the authors meaning?? there are compounds produced from LAB such as lactic acid or bacteriocins able to avoid Salmonella growth...

Reviewer 3 Report

    In this study, 89 strains of Lactobacillus from chicken intestines were isolated by the national standard method, Gram staining, physiological and biochemical experiments, and molecular sequencing; The inhibition characteristics of 89 strains of chicken derived Lactobacillus against 10 strains of Salmonella were detected by agar inhibition zone, The results showed that the inhibition zone of 24 strains of chicken derived Lactobacillus was more than 10 mm, which indicated that the isolated chicken derived Lactobacillus could effectively inhibit the growth of Salmonella. 

There are a lot of comments for the authors to be addressed before the final acceptance. A lot of sections in this manuscript need to be re-written scientifically as the quality of the manuscript writing is very bad especially for the results section. There are sets of good data in this manuscript but the presentation is very bad. Therefore, I strongly suggest that the authors read articles on a similar topic to see how to present their data. Here are some comments to improve the manuscript

1- In abstract: please add the name of the used salmonella strain

2- In the introduction, please add a paragraph about the economic importance of salmonella

3- line 51: change inhibition to the efficacy

4- line 51: please add the name of the used salmonella strain

5- line 59: please add a reference

6- line 60: serially diluted!

7- please add the source of your chemicals and cells throughout the materials and methods section (company, state, country)

8- line 65: what kind of salmonella strains? please provide names

9- line 57: change the title to (strain isolation, culture media, and used cell lines)

10- line 67: please describe the maintaining media of Caco-2 cells with references

11- Is gram stain is accurate enough to confirm the isolated strains. please explain and add a reference

12- line 68: please provide references for this technique

13- line 83: Do you mean 100 μL from all strains together or from each strain

14- line 86: please add the number of the bacteria in CFU

15- please rephrase from line 87-90

16- please add references to the drug resistance technique

17- please add references for the bile salt tolerance test

18- please add references for the Adhesion of different bacteria to Caco-2 cells

19- number of the bacteria should be presented as CFU/ml

20- please combine all 2.6 subtitles as one section and try to avoid redundancy with emphasis that lactobacillus was used as treatment and as prophylactic and highlight the difference.

21- please add two sentences at the beginning of each experiment to explain the aim of each experiment

22- add references for section 2.7

23- line 155: what kind of cell samples do you mean? and how much ml did you use?

24- line 161: change vibration to shaking

25- section 2.9 need to move to the beginning, directly after section 2.1 

26- your materials and methods section lacked references. please also the number of repeats for each experiment

27- in each experiment: what controls did you use?

28- where is the statistical analysis section?

29- line 178: add primers concentration

30- line 183: what is subsection?

31- you need to rephrase all the results section again. you will need to write/ explain the results then refer to the respective figure, not the opposite. Please stop just describing the figures through the result section

32- line 184: it is better to present the inhibition results as a percentage compared to the control

33- line 193-194: please add the names of these antibiotics

34- line 82-91: please add more details on the screened antibiotics and their full names with abbreviations, MIC, and the cut off for resistance and susceptibility. why you specifically selected these antibiotics?

35- line 190-195, what about MDR

36- you need to add your statistical analysis in the result section and the p-value

37- line 197-205: please explain your results rather than describe the figures 

38- figures 2 and 3 are not clear

39- line 109: which control did you use for this comparison

40- line 208-212: you need to present your results as log CFU or inhibition%. You also need to talk about your control

41- line 213-224: this paragraph is confusing

42- You have a lot of abbreviations without a full name. please add the full name to the abbreviations  the first time you mention it in the manuscript

43- the name of the bacteria should be in italic

44- section 3.1.6: please divide into different subtitles and follow similar comments as the other sections

45- figure S4 is not complete

46- please put the table and the figures in the respective places in the manuscript

47- please move the biochemical testes and characterization materials and methods and results to the beginning after sampling

47- line 552-553: I assumed that you know the source of your isolates and their history. Aren't you?

48- It will be very interesting to test these lactobacilli strains in Salmonella infected chickens

49- your discussion is very brief. Please expand and discuss all the obtained results in your study and compare them to the previous studies

50- You used a very low number of references to support your sudy